# Platelets at the Crossroads of Pro-Inflammatory and Resolution Pathways during Inflammation

**DOI:** 10.3390/cells11121957

**Published:** 2022-06-17

**Authors:** Nadine Ludwig, Annika Hilger, Alexander Zarbock, Jan Rossaint

**Affiliations:** Department of Anesthesiology, Intensive Care and Pain Medicine, University Hospital Muenster, 48149 Muenster, Germany; n_ludw02@uni-muenster.de (N.L.); annika.hr@gmx.de (A.H.); zarbock@uni-muenster.de (A.Z.)

**Keywords:** platelets, platelet-leukocyte interactions, inflammation, resolution, pulmonary inflammation, COVID-19

## Abstract

Platelets are among the most abundant cells in the mammalian circulation. Classical platelet functions in hemostasis and wound healing have been intensively explored and are generally accepted. During the past decades, the research focus broadened towards their participation in immune-modulatory events, including pro-inflammatory and, more recently, inflammatory resolution processes. Platelets are equipped with a variety of abilities enabling active participation in immunological processes. Toll-like receptors mediate the recognition of pathogens, while the release of granule contents and microvesicles promotes direct pathogen defense and an interaction with leukocytes. Platelets communicate and physically interact with neutrophils, monocytes and a subset of lymphocytes via soluble mediators and surface adhesion receptors. This interaction promotes leukocyte recruitment, migration and extravasation, as well as the initiation of effector functions, such as the release of extracellular traps by neutrophils. Platelet-derived prostaglandin E2, C-type lectin-like receptor 2 and transforming growth factor β modulate inflammatory resolution processes by promoting the synthesis of pro-resolving mediators while reducing pro-inflammatory ones. Furthermore, platelets promote the differentiation of CD4^+^ T cells in T helper and regulatory T cells, which affects macrophage polarization. These abilities make platelets key players in inflammatory diseases such as pneumonia and the acute respiratory distress syndrome, including the pandemic coronavirus disease 2019. This review focuses on recent findings in platelet-mediated immunity during acute inflammation.

## 1. Introduction

When Max Schultze and Giulio Bizzozero first described platelets as constituents of human blood and discovered their participation in wound closure in the second half of the 19th century, they might have already guessed the potential of these tiny, abundant cells [1]. Since then, platelets have been widely investigated in numerous research projects with a recently expanding focus from traditional coagulation and wound healing studies toward their role in inflammation and inflammatory resolution processes.

Platelets are small, anucleate cells in the mammalian blood circulation. They are released from megakaryocytes in a process called thrombopoiesis, which is mainly driven by the acidic glycoprotein (GP) thrombopoietin (TPO) [2]. While the bone marrow remains the major site of platelet biogenesis, populations of megakaryocytes, both mature and immature, were recently found in the lungs’ extravascular space [3]. Interestingly, lung megakaryocytes express vast amounts of toll-like receptor (TLR)2, TLR4 and major histocompatibility complex (MHC)II, indicating that they are primed towards an immune-specific phenotype [4].

The final platelet concentration in human blood varies between 150 and 450 × 10^9^ platelets per liter, which is maintained by a production and clearance of roughly 10^11^ platelets per day [5]. Platelets remain in the bloodstream for 7 to 10 days and are prepared to carry out different tasks in hemostasis and immunity upon stimulation [6]. They sense and respond to a wide range of signals from the vascular endothelium and circulating blood cells by surface receptor stimulation and extensive signaling events [7]. Platelet activation results in morphological changes driven by the reorganization of the cortical actin cytoskeleton from an initially disk-shaped form to a considerably enlarged fully spread platelet [8]. The physical interaction of platelets with endothelial cells, leukocytes and extracellular matrix components is mediated by surface integrins. Platelets primarily express β_1_ and β_3_ integrins: for instance, α_2_β_1_ (GPIa/IIa) and α_IIb_β_3_ (GPIIb/IIIa). The activation of platelets induces conformational changes of α- and β-subunits, thus enabling ligand binding and subsequent intracellular signaling cascades [9,10]. Since de novo protein expression in anucleate platelets is limited by the presence of only a selection of different mRNA copies, protein conversions trigger procoagulant activation, aggregation, and secretion of granule components [11]. The three major types of platelet granules—α-granules, dense granules, and lysosomal granules—are packaged with a variety of different proteins and substances capable of influencing both coagulation and inflammatory processes. 

In the context of inflammation, platelets were found to capture pathogens and release antimicrobial substances. Furthermore, they promote not only pro-inflammatory but also inflammatory resolution processes by releasing a variety of growth factors, cytokines and chemokines and by actively interacting with neutrophils, monocytes and lymphocytes [12]. In biomedical research, the participation of platelets in immune responses is of great interest to develop novel treatment strategies for clinical use. In particular, since anti-platelet medication has previously been associated with modified immunity [13]. The coronavirus disease 2019 (COVID-19) pandemic is a prime example for ongoing intensive research on platelet behavior during severe inflammation.

This review article will provide an overview about recent findings exploring the immune modulatory functions of platelets, with a special focus on platelet-leukocyte interactions during acute inflammation and resolution in respiratory diseases.

## 2. Immune-Modulatory Function of Platelets

The rheological properties of blood lead to a distinct distribution of circulating cells within the vessels. Larger cells, including erythrocytes and leukocytes are located centered, while small platelets are enriched closer to the endothelial cell surface [10]. Thus, platelets patrol the blood in close proximity to the vessel walls for vascular injury or insult. Upon tissue trauma, they support hemostasis as their primary physiological role. Platelets are further among the cells constituting the first line of defense against all kinds of intruders in the circulation. They restrict the dissemination of pathogens by occluding vessel walls, exercising direct pathogen capture and promoting subsequent immune responses [14]. Moreover, platelets directly interact with various bacterial strains, including *Staphylococci*, *Streptococci* and *Klebsiella pneumoniae* [7,15,16,17]. To prevent bacterial spreading, platelet GPIIb/IIIa and Fcγ receptors facilitate adherence and aggregation by installing fibrinogen and fibronectin around bacteria, thereby forming platelet–pathogen complexes [18]. However, it has to be taken into account that pathogens were also widely described to intentionally attack platelets during infections to cause altered platelet functions and counts [19]. As an example, *Helicobacter pylori* and *Staphylococcus aureus* promote thrombus formation during infections by facilitating platelet binding via immobilized von Willebrand Factor (vWF) [20,21]. Furthermore, Kerrigan and colleagues found platelet GPIb to be the major receptor used by *Streptococcus sanguis* to induce platelet aggregation with fundamental consequences for the pathogenesis of infective endocarditis [16].

### 2.1. Platelet Toll-like Receptors

The rapid detection of microbial infections by the mammalian innate immune system particularly depends on the expression of Toll-like receptors (TLRs). TLRs are pathogen-associated molecular pattern (PAMP) and danger-associated molecular pattern (DAMP) recognition receptors and facilitate innate immune responses to foreign organisms [22]. The stimulation of TLRs results in the specific transcription of antimicrobial genes via a nuclear translocation of nuclear factor-κB (NF-κB), which promotes the subsequent production of NF-κB-controlled pro-inflammatory cytokines such as interleukin (IL)-1, IL-6 and IL-8 [23]. Furthermore, the initiation of the innate immune response highly influences a successive antigen-specific response by the adaptive immune system [24].

Until now, 10 human TLRs have been identified, which are all expressed on the cells of the innate immune system, including neutrophils, basophils, eosinophils, monocytes, macrophages, dendritic cells, natural killer (NK) cells and mast cells [25]. Interestingly, mRNA transcripts of all 10 TLRs have been recognized in human platelets, indicating their prominent role for adequate innate immune responses. Extensive studies have proposed the functional protein expression of TLR1, TLR2, TLR3, TLR4, TLR6, TLR7 and TLR9 in platelets (Figure 1) [26,27,28,29,30,31].

In 2010, Spinelli et al. verified the presence of functional NF-κB in platelets, thus supporting a non-classical role for the NF-κB family in anucleate cells [32]. This finding was further proven by Rivadeneyra and colleagues by demonstrating that TLR2 and TLR4 triggered platelet activation induces IκBα degradation and p65 phosphorylation [33]. TLR2 and TLR4 engage components of Gram-negative bacteria and are expressed more abundantly in platelets compared to other TLRs [31].

TLR2 is primarily expressed on the platelet surface and is considered a unique TLR since it heterodimerizes with either TLR1 or TLR6 to recognize numerous, different bacterial lipopeptide structures [34]. The stimulation of TLR2/TLR1 via synthetic Pam3CSK4 increases P-selectin (CD62P) surface mobilization, activates GPIIbIIIa and eventually causes platelet aggregation. Moreover, TLR2/TLR1-induced platelet activation is connected to phosphoinositide 3-kinase (PI3K)/protein kinase B (Akt) signaling [35]. Damien et al. highlighted an activation of extracellular-signal regulated kinases (ERK)1/2 and p38 mitogen-activated protein kinase (p38 MAPK) upstream of Akt phosphorylation and additional slow phosphorylation of NF-κB upon TLR2-stimulation [36]. In a recent study investigating the role of platelet TLR2/TLR6 in inflammation, the TLR2/TLR6 agonist Pam2CSK4 induced platelet granule secretion and GPIIbIIIa activation as well as platelet adhesion to collagen-coated surfaces and to human umbilical vein endothelial cells by involving TLR2/NF-κB and Bruton’s tyrosine kinase (BTK) signaling [37].

Functional TLR4 on platelets was detected in 2005 in a study by Andonegui et al. Their experiments indicated a role for TLR4 in murine platelets for fibrinogen binding under flow conditions; however, CD62P expression was not altered upon TLR4 activation. Furthermore, lipopolysaccharide (LPS), the major TLR4 ligand, induced thrombocytopenia in wildtype but not in TLR4-deficient mice [30]. At a later time point, the classical components of the LPS receptor-signaling complex—CD14, MD2 and MyD88—were found in platelets and associated with LPS-mediated potentiation of platelet aggregation [38]. Beyond that, platelet TLR4 has been widely investigated in the context of classical inflammatory diseases, such as sepsis, pneumonia and Crohn’s disease [39,40,41]. In a murine model of endotoxemia, Stark et al. demonstrated that platelet TLR4 promotes microvascular thrombosis. This suggestion was verified by transfusing TLR4 deficient mice with wildtype platelets and analyzing responsiveness to LPS with regards to microvascular thrombosis by intravital microscopy [42]. More recent evidence suggests that sepsis affects neither platelet TLR4 expression nor classical platelet activation, although platelets did respond to LPS by an increase in mitochondrial respiration [43]. Certainly, the response of platelets to LPS as well as TLR4-induced interaction of platelets with leukocytes is still a matter of debate [31].

### 2.2. Platelet Granules

Upon stimulation, platelets release their granule contents into the circulation. Granule exocytosis is initiated by surface receptor activation, leading to an increase in intracellular calcium levels and the activation of phosphokinase C (PKC) [44]. Granule-plasma membrane fusion and content release are primarily driven by a soluble N-ethylmaleimide-sensitive factor attachment protein receptor (SNARE)-dependent mechanism [45]. Released contents vary depending on the respective granule type.

With a size of 200–500 nm in diameter and 50–80 per platelet, α-granules are the largest and most prominent platelet granules [46]. They are the major storage granules of platelets and are sorted and packaged differentially by megakaryocytes during maturation, thereby providing different α-granule subpopulations [47]. Platelet α-granules contain hemostatic and (anti-)angiogenic substances, growth factors, proteases, necrotic factors, immunoglobulins, adhesive glycoproteins, and a vast number of cytokines and chemokines [46]. Among others, platelet factor (PF)4, CD62P, CD40L, regulated and normal T cell expressed and secreted (RANTES), transforming growth factor (TGF)-β and macrophage inflammatory protein (MIP)-1α are prime examples for α-granule-derived inflammatory mediators. Their release may affect leukocyte recruitment and adhesion, as well as cell proliferation and differentiation [48,49,50,51]. Moreover, platelet α-granules store a variety of peptides with microbicidal activity such as connective tissue-activating peptide (CTAP)-3, thymosin β-4 and fibrinopeptide A and B [52]. In a study from Krijgsveld et al. α-granule-derived thrombocidin (TC)-1 and -2 were able to prevent the growth of bacteria such as *Bacillus subtilis*, *Escherichia coli*, *Staphylococcus aureus* and even fungi such as *Cryptococcus neoformans* [53].

Dense granules (δ-granules) constitute the second most abundant platelet granules with 3–8 granules per platelet. They measure about 200 nm in diameter and mainly contain bioactive amines (serotonin and histamine), bivalent cations and nucleotides (ADP, ATP, GDP and GTP). These components influence not only leukocyte functions and trafficking but also endothelial cell reactivity [54,55,56]. The large amounts of comprised calcium provides δ-granules with an electron-dense property, allowing for direct visualization in electron microscopy [11].

Acid proteases and glycohydrolases enable lysosomal granules to fulfill their functions in digesting phagocytic and cytosolic components. Furthermore, the release of lysosomal contents supports the degradation of extracellular matrix components, fibrinolysis and receptor cleavage [57].

In a study investigating platelet TLR9, Thon et al. found a novel electron-dense tubular system-related compartment which they named “T granules”. The authors discovered TLR9 colocalization with protein disulfid isomerase and vesicle-associated membrane protein (VAMP)7 or VAMP8 in human platelets, thus characterizing this fourth type of platelet granule [58]. However, T granules remain poorly understood and are a matter of ongoing discussion.

### 2.3. Platelet-Derived Microvesicles

An increasing number of studies emphasized a growing interest in the functions of platelet-related extracellular vesicles (EVs). EVs comprise a number of different vesicles consisting of phospholipid bilayers with a diameter ranging from 30 to 3000 nm. Depending on their cellular origin, they contain different proteins, lipids and even nucleic acids [59]. Currently, three types of EVs are known, including exosomes (30–150 nm), microvesicles/microparticles (MVs/MPs, 100–1000 nm) and apoptotic bodies (1000–3000 nm). MVs are released from several cell types such as leukocytes, endothelial cells and platelets, and they appear in different biological fluids [60]. They had long been considered a machinery for cellular maintenance before their purpose as a sophisticated communication system between local and distant cells has been discovered. Since then, MVs have been recognized as promising biomarkers in infectious diseases and cancer [61]. Ultracentrifugation became a successful tool to isolate MVs from peripheral blood and subsequent flow cytometric analyses allowed for the specific discrimination of MV subpopulations [62]. Platelet-derived MVs (PMVs) are the most abundant MV type in the circulation [63]. They are released via surface shedding upon loss of cytoskeleton-membrane adhesion and exhibit classical platelet-related surface markers, such as GPIIbIIIa, GPIbα and CD62P [64]. It is generally accepted that PMVs act as pro-inflammatory mediators in inflammatory diseases. Gkaliagkousi et al. found a significant increase in PMV concentrations in patients with newly diagnosed type 2 diabetes. Elevated PMV levels were associated with glycaemic profiles and amplified thrombotic tendencies [65]. Similar observations were reported by Bratseth and colleagues, highlighting elevated PMV levels in coronary artery disease patients with type 2 diabetes and albuminuria as potential predictors of disease severity [66]. In a project investigating the influence of PMVs in early diabetic nephropathy Zhang et al. revealed that PMVs induce the production of reactive oxygen species (ROS), decrease nitric oxide levels and promote barrier permeability in glomerular endothelial cells. Additionally, C-X-C chemokine ligand (CXCL)7 originating from PMVs affected glomerular endothelial injury via the G protein-coupled receptors C-X-C chemokine receptor (CXCR)1 and CXCR2 [67].

Boilard et al. performed extensive studies by analyzing PMV release and phenotypes during rheumatoid arthritis. Synovial fluids of patients suffering from rheumatoid arthritis showed tremendous clusters of PMVs. In vitro experiments indicated that platelets release MVs in a collagen-GPVI-dependent process triggered by an interaction of platelets with synoviocytes and their extracellular matrix. PMVs further caused secretion of the pro-inflammatory cytokines IL-6 and IL-8 from fibroblast-like synoviocytes upon IL-1-mediated activation [68]. In a recent study assessing the clinical significance of PMVs in acute pancreatitis, Qi and colleagues described a positive correlation in disease severity with elevated PMV levels in patient plasma. Moreover, the authors reported that PMVs derived from patients with acute pancreatitis induced the formation of extracellular traps (NETs) in neutrophils from healthy volunteers. NETs are extracellular, web-like chromatin fibres equipped with cytosolic and granule proteins, such as myeloperoxidase (MPO) and neutrophil elastase (NE), that trap and neutralize pathogens. However, an uncontrolled and excessive release of NETs may cause damage to healthy host tissues [69]. Thus, PMVs might promote disease progression in acute pancreatitis by affecting the formation of pro-inflammatory and potentially harmful neutrophil-derived mediators [70].

During severe trauma and sepsis, activated platelets present enhanced PMV shedding [71]. In a murine model of caecal ligation and puncture (CLP)-induced systemic inflammation, a threefold increase in PMV levels compared to the control animals was observed. The same study further revealed the influence of PMVs on platelet-dependent thrombin formation via PS exposure and, thus, stressed the prominent role of PMVs in the development of dysfunctional coagulation under septic conditions [72]. Janiszewski et al. demonstrated that PMVs from septic patients possess subunits of the NADPH oxidase and exhibit intrinsic ROS production, thereby increasing apoptosis rates in endothelial and vascular smooth muscle cells [73]. Controversially, a later study investigating early inflammatory responses in patients suffering from severe sepsis suggested a positive impact in increased PMV and endothelial-derived MV levels on organ damage and survival. These results contradicted the common assumption that excessive and uncontrolled immune processes promoted the pathogenesis of sepsis [74]. Interestingly, recent evidence supports the protective role of PMVs during septic conditions. Platelets were demonstrated to transfer functional microRNA (miRNA) miR-223 via PMVs into endothelial cells. MiRNAs represent an intercellular communication system for regulating the expression of target genes [75]. The uptake of miR-223-containing septic PMVs resulted in a downregulation of endothelial intercellular adhesion molecule (ICAM)-1 and reduced binding of peripheral blood mononuclear cells, suggesting a PMV-mediated reduction in leukocyte migration [76].

Elevated plasma levels of PMVs have also been observed in association with cancer progression. Platelet-derived properties of PMVs to interact with cancer cells via surface proteins enhanced platelet-mediated masking and attraction of tumor cells, stability of putative nidation sites, interaction of tumor cells with endothelial cells and extravasation of tumor cells [77]. PMVs are massively increased in patients with gastric cancer and depict the highest potential as diagnostic biomarkers in comparison with vascular endothelial growth factor, IL-6 and RANTES [78]. Studies investigating oral squamous cell carcinoma and breast cancer further present the effect of PMVs to enhance procoagulant activities and the invasive character of cancer, respectively [79,80].

These findings highlight the diverse functions of PMV and their enormous potential for therapeutic intervention.

### 2.4. Platelet-Leukocyte Interactions

The interplay between platelets and leukocytes emphasizes the immunomodulatory functions of platelets like no other process. Communication with neutrophils, monocytes and a subset of lymphocytes allows platelets to actively modulate innate and adaptive immune responses. The interaction relies on a transfer of biochemical signals through adhesive receptors and an exchange of soluble mediators (Figure 2) [81].

Physical contact between platelets and leukocytes is initiated by interaction of platelet CD62P and P-selectin glycoprotein ligand (PSGL)-1 on leukocytes [82]. Subsequent CD11b/CD18 (macrophage-1 antigen, Mac-1) binding to platelet GPIb or Mac-1 binding to GPIIbIIIa via fibrinogen enables firmer adhesion [83,84]. The formation of platelet-leukocyte aggregates mediates outside-in signaling events and, thus, facilitates leukocyte migration and function [10]. Leukocytes may bind a varying number of platelets and complexes may appear either unbound in the circulation or bound to vessel walls and tissues [85]. Flow cytometry, tissue section imaging and live cell imaging represent the most common tools to analyze platelet leukocyte aggregates by applying specific fluorescent antibodies to identify different cell populations [86]. A growing body of literature has examined platelet–leukocyte interactions under disease conditions, indicating their prominent role for a comprehensive immune response. In addition, platelet–leukocyte aggregates have been suggested as potential biomarkers in various pathologies including pulmonary diseases, vasculitis and cardiovascular diseases [85].

In 2010, Kornerup and colleagues discovered that neutrophil recruitment for inflammatory sites strongly depends on the expression of PSGL-1 and the presence of platelets. Further results suggested a “chaperon”-like role of platelets for neutrophil recruitment [87]. Interestingly, recruited neutrophils scan the bloodstream for platelets before they proceed their function during infections. Within activated venules, neutrophil polarization induces a protruding domain to contact platelets. The following signal transduction via PSGL-1 provokes receptor redistributions that are essential for subsequent neutrophil migration [88]. A study from 2016 revealed that the interplay of platelets and neutrophils involves a complex multistep reciprocal crosstalk. CD62P allows platelets to adhere to intravascular neutrophils, thereby mediating GPIbα-induced neutrophil EV formation. Subsequently, neutrophil-derived arachidonic acid is transferred via EV exocytosis into platelets and processed to synthesize thromboxane A2 (TxA2). Platelet-derived TxA2, in turn, drives the endothelial expression of ICAM-1 and, thus, promotes full neutrophil activation and an adequate immune response [89]. Platelets additionally mediate neutrophil and monocyte diapedesis through venular microvessels via CD40-CD40L-dependent interactions at endothelial junctions. CD62P-PSGL1 binding causes conformational changes of leukocyte integrins via ERK1/2 MAPK signaling and promotes transmigration involving ICAM-1 and -2, vascular cell adhesion molecule (VCAM), platelet endothelial cell adhesion molecule (PECAM) and junctional adhesion molecule (JAM)-A [90]. A recent study analyzed the impact of platelet-specific PI3K on the formation of neutrophil-platelet and monocyte-platelet aggregates in a murine model of acid-induced lung injury. PI3K was identified as crucial for cell–cell interactions and protein deficiency prevented leukocyte extravasation into the bronchoalveolar space [91]. These results augment earlier findings that the inhibition of platelet-neutrophil aggregation in acid-induced lung injury improved gas exchange and reduced vascular permeability, thus allowing the prolonged survival of experimental animals [92]. In addition, it was indicated that the formation of monocyte-platelet aggregates shapes the phenotype of monocytes toward a pro-inflammatory direction depending on the degree of platelet activation [93]. Platelets were also recognized to modulate lymphocyte function by supporting lymphocyte rolling, adhesion and migration to inflamed sites [94]. Li et al. specified that approximately 3% of circulating lymphocytes present bound platelets under normal conditions. Lymphocyte activation led to increased platelet binding involving all classical platelet–leukocyte receptors such as CD62P, GPIIb/IIIa, Mac-1 and CD40. Furthermore, platelets bound rather large, activated T-lymphocytes than smaller B-cells [95]. Interestingly, sepsis-relevant bacterial pathogens, such as *Klebsiella pneumoniae* and *Staphylococcus aureus*, enhanced the formation of aggregates, suggesting a prominent role of platelet–leukocyte interactions during systemic inflammation [96]. In a small patient study by Gawaz et al., the levels of circulating platelet-leukocyte aggregates were increased under septic conditions. In non-survivors, aggregates were less abundant and associated with enhanced peripheral sequestration [97]. Moreover, platelet-monocyte aggregates were demonstrated to positively correlate with 28-days mortality in older (≥65 years) patients and might serve as predictors for disease severity [98].

As previously indicated, platelet-leukocyte interactions are not only mediated by physical contact but especially by a transfer of soluble mediators between cells. Platelet-derived PF4 specifically binds to neutrophils and elicits firm adhesion to endothelial cells in a dose-dependent manner involving L-selectin (CD62L) and lymphocyte function-associated antigen 1 (LFA-1) [99]. Furthermore, PF4 promotes monocyte chemotaxis via CC chemokine receptor (CCR)-1 [100]. Platelet-released RANTES attracts human eosinophils and triggers shear-resistant monocyte arrest on inflamed endothelial cells [101,102]. Interestingly, PF4 and RANTES form heterodimers and amplify RANTES-triggered effects on monocytes and monocytic cells emphasized by enhanced arrests on activated endothelial cells [103]. Similarly to PF4, CXCL7 depicts structural properties allowing the formation of biologically active homo- and heterodimers. Receptor activity is regulated by binding of sulfated glycosaminoglycans [104]. A CXCL7 chemotactic gradient within thrombus bodies attracts leukocytes to the sites of endothelial damage and promotes leukocyte adhesion [105,106]. In addition to a range of chemokines and cytokines, platelets secrete DAMPs such as high-mobility group box 1 (HMGB1)—a nonhistone, DNA-binding nuclear protein. Activated platelets present HMGB1 at the cell’s surface and release it into the circulation. Platelet-derived HMGB1 is a well-known mediator of thrombosis [107]. Furthermore, HMGB1 from platelets regulates monocyte recruitment and apoptosis via the receptor for advanced glycation end products (RAGE) and TLR4, respectively [108].

There is a considerable amount of literature on platelets mediating the formation of NETs. In 2007, Clark et al. demonstrated that high-dose LPS stimulation results in platelet binding to neutrophils and a subsequent release of NETs under in vitro shear stress. These processes were also observed in a murine model of endotoxemia, highlighting NET formation upon platelet TLR4 activation in liver sinusoids and pulmonary capillaries as an effective, threshold-induced mechanism for bacterial trapping [109]. Platelets induce NET formation in transfusion-related acute lung injury (TRALI) involving Raf/MEK/ERK signaling. Here, NETs disturb the integrity of endothelial monolayers and, thus, contribute to a harmful disease progression [110]. In a murine model of ventilator-induced lung injury, Rossaint et al. identified synchronized integrin-mediated engagement and heterodimerization of platelet-derived PF4 and RANTES as crucial for platelet-induced NET formation [111]. This is in line with findings in patients with myocardial infarction, demonstrating that the prevention of PF4-RANTES interaction via treatment with the cyclic peptide MKEY reduces NET formation following myocardial ischemia/reperfusion injury [112].

### 2.5. Platelets in Inflammatory Resolution

The pathology of many diseases is driven by a dysregulated and persistent state of inflammation, which is observable for instance in the development of the acute respiratory distress syndrome (ARDS) and sepsis. Therefore, tightly regulated, active processes promoting the resolution of acute inflammation are essential for preventing needless and prolonged host tissue damage [113]. Resolution is driven by spatial and temporal production of pro-resolving mediators. Classical pro-resolution mediators comprise lipoxins, resolvins, protectins and maresins, annexin A1, hydrogen sulfide and carbon monoxide, as well as special neurotransmitters and peptides [114]. Fundamental processes for the resolution of inflammatory events include the termination of cell extravasation, counter-regulation of pro-inflammatory mediators, reprogramming of macrophages, induction of apoptosis and removal of leukocytes especially neutrophils, as well as the initiation of wound healing. These mechanisms eventually result in the clearance of pro-inflammatory cells and the restoration of functional homeostasis [115].

Regulating platelet function is crucial for a steady resolution of inflammation. In various cardiovascular diseases, cyclooxygenase (COX) and P2Y12 inhibitions were used to dampen platelet functions and impeded thromboembolic events. Aspirin prevents the COX-1- and COX-2-dependent production of TxA2, while P2Y12 receptor antagonists such as clopidogrel impaired platelet ADP signaling [116]. Interestingly, the application of aspirin and clopidogrel not only suppressed thrombotic platelet functions but also platelet-mediated pro-inflammatory processes including the emergence of platelet-leukocyte aggregates, platelet-mediated leukocyte recruitment and ROS formation [117,118,119]. Liverani et al. investigated the role of P2Y12 in neutrophil migration and lung inflammation by using P2Y12 null mice in a model of acute lung injury and sepsis. Platelet activation and platelet-leukocyte aggregates were diminished, which was associated with organ protection. Similar results were obtained upon clopidogrel treatment [120]. Furthermore, clopidogrel was recently used to successfully inhibit platelet inflammasome assembly and, thus, the release of pro-inflammatory IL-1β and IL-18 under septic conditions resulting in improved renal function [121]. This underlines the importance of regulating platelet activity for the initiation of anti-inflammatory processes.

Platelets were found to directly influence anti-inflammatory and resolution processes (Figure 3) [12]. Abdulnour et al. demonstrated that platelets participate in the biosynthesis of the pro-resolving protein maresin 1 via neutrophil interactions. The production of maresin 1 is induced by platelet-derived lipoxygenase (12-LOX), which converts docosahexaenoic acid to 13S,14S-epoxy-maresin. Further protein processing by neutrophils eventually results in full-length maresin 1. In a murine model of ARDS, intravascular maresin 1 decreased neutrophil counts in the lung and was clearly organ-protective [122]. Interestingly, maresin 1 enhances ADP and thrombin-induced haemostatic functions of platelets such as aggregation, spreading and δ-granule release. However, the release of the inflammatory mediators TxB2, prostaglandin E2 (PGE2), CD62P and PMVs from thrombin-activated platelets was significantly reduced by maresin 1 supplementation [123]. Resolvin E1 is another pro-resolving mediator influencing platelet function. ADP-induced actin polymerization and CD62P surface mobilization are inhibited by resolvin E1. Furthermore, resolvin E1 blocks ADP-mediated platelet aggregation by regulating P2Y12 signaling [124].

In a study from 2017, platelets released high amounts of PGE2, which increased the production of the anti-inflammatory cytokine IL-10 by monocytes/macrophages via PGE2 receptors EP2 and EP4. IL-10, in turn, prevented the release of monocyte/macrophage-derived TNFα. Thus, platelets initiated a cross-regulation in monocytes and macrophages to inhibit pro-inflammatory processes [125]. The importance of the podoplanin receptor C-type-lectin-like-2 (CLEC-2) on platelets, which mitigates inflammation in sepsis, was demonstrated in a murine model of systemic inflammation. The deletion of CLEC-2 on platelets was associated with impaired vascular integrity in the peritoneum, as well as decreased liver and kidney function. The interaction of CLEC-2 on platelets with podoplanin on macrophages presents an anti-inflammatory axis and regulates immune cell infiltration during sepsis [126]. Interestingly, GPIb-IX is an additional platelet receptor associated with anti-inflammatory properties during polymicrobial sepsis. The lack of GPIb-IX results in reduced platelet-leukocyte aggregates but increased levels of pro-inflammatory cytokines IL-6, CXCL1 and monocyte-chemoattractant protein (MCP)-1 [127]. Platelets further enhance T helper (Th)1, Th17 and regulatory T cell (Treg) differentiation of CD4^+^ T cells via cell–cell contacts and release of PF4, RANTES and TGFβ. Furthermore, the distinct regulation of CD4 effector response by platelets is achieved by the TGFβ-mediated inhibition of forkhead box (FOX)P3 T cell proliferation [128].

A recent study from Rossaint et al. highlighted the impact of platelets in the resolution of pulmonary inflammation. In a murine model of bacterial pneumonia, platelet depletion resulted in a decrease in neutrophil apoptosis and, thereby, promoted persistent high neutrophil counts along with eventual fibrotic tissue remodeling. The role of platelets relied on a sophisticated interplay with Tregs promoting inflammatory resolution by the release of IL-10 and TGFβ. Physical Treg-platelet interaction was mediated by sCD40L and CD62P-PSGL-1 binding and resulted in remodeling the Treg transcriptome. Extensive in vitro assays and macrophage transcriptome analyses revealed that the interaction of platelets with Tregs is essential for driving macrophages towards an anti-inflammatory and pro-resolving fate during pulmonary inflammation [129].

## 3. The Role of Platelets in Pulmonary Inflammation

ARDS is a severe condition of acute and persistent lung inflammation and a common cause of respiratory failure in critically ill patients. This life-threatening condition is characterised by alveolar damage, disruption of the endothelial-epithelial barrier, increased permeability, formation of pulmonary edema and decreased pulmonary gas exchange [130]. In 2012, clinical diagnostic criteria for ARDS were updated and specified in the “Berlin definition of ARDS in adults” comprising specifications for timing and origin of respiratory failure, lung imaging via chest radiograph or computer tomography and conditions for oxygenation [131]. The most common conditions associated with ARDS include pneumonia, non-pulmonary sepsis, major trauma and aspiration of gastric contents. *Staphylococci*, *Klebsiella*, *Streptococci*, influenza viruses and coronaviruses are among the most prominent bacterial and viral pathogens causing nosocomial and community-acquired pneumonia [132]. The pathogenesis of pneumonia-induced ARDS is multi-factorial and involves not only pathogenic insults but also excessive host immune responses associated with an aberrant release of cytokines and proteolytic enzymes mediating the propagation of lung injury [133]. Various studies demonstrated a crucial impact of platelets during bacterial- and viral-induced pulmonary inflammation and lung injury.

In a recent trial by Cleary et al. mice were exposed to LPS vapor, which resulted in neutrophil and platelet recruitment into the lung vasculature. Interestingly, platelet recruitment was CD62P- and PSGL-1-independent and was not affected by neutrophil depletion, thereby emphasizing a major role for platelets in host defense during lung inflammation [134]. However, platelet and endothelial CD62P are essential for an adequate immune response against *Klebsiella pneumoniae*. Although leukocyte recruitment was unaffected, bacterial loads in the lung and blood of CD62P knockout mice were significantly enhanced upon *Klebsiella*-induced pneumonia and were associated with distant organ damage. In this context, the manipulation of CD62P function was suggested to improve the outcome of pneumosepsis [135]. De Stoppelaar and colleagues identified that platelet activation by *Streptococcus pneumomiae* is independent of TLR2, TLR4 and TLR9 signaling [136]. Interestingly, in a recent study, *Streptococcus pneumoniae*-derived pneumolysin initiated the mobilization of CD62P from platelet α-granules and the release of platelet and neutrophil EVs. EVs from neutrophils also influenced platelet activity, thereby modulating platelet functions during pneumococcal infection [137]. In methicillin-resistant *Staphylococcus aureus* infections, platelets and their releases enhanced bacterial uptake and killing by dendritic cells in a CD40L-dependent manner. The expression of CD80, TNFα, IL-12 and IL-6 is upregulated in dendritic cells after contact with activated platelets, demonstrating the role of platelets for the induction of the adaptive immune response upon *Staphylococcus aureus* infection [138]. Platelets are known to maintain the basal alveolar–capillary barrier during infection pneumonia. In a murine model investigating pulmonary *Pseudomonas aeruginosa*-infection, Mpl knockout elicited alveolar barrier disruption and haemorrhagic pneumonia. Platelet reconstitution rescued the induction of lung injury by protecting against secreted bacterial T2SS exotoxins [139].

Enhanced platelet activation upon pulmonary inflammation results in the deposition of platelets within the pulmonary microvasculature and subsequent thrombocytopenia, which is strongly associated with the development of ARDS [140]. In this context, platelets may mediate vascular leakage and increase alveolar-capillary permeability due to impaired endothelial cadherin stability [141]. Leukocyte–platelet aggregate formation during ARDS affects intercellular signaling and the release of inflammatory mediators, thereby altering the inflammatory milieu [142]. NO inhalation therapy in patients suffering from ARDS mitigates the formation of platelet-leukocyte aggregates by decreasing CD62P mobilization via cGMP-PKG signaling [143,144].

During influenza pneumonia platelets accumulate in infected lungs and form aggregates with neutrophils. Aggregate formation triggers the release of NETs that contribute to alveolar–capillary damage [145]. Not only platelet–neutrophil but also platelet–endothelial interactions contribute to lung injury during influenza infection. Upon influenza stimulation, endothelial cells present elevated expression of platelet receptor ligands vWF and ICAM-1. However, platelet adhesion to endothelial cells is mediated by integrin α_5_β_1_ and fibronectin deposition. In this context, platelet adhesion to lung endothelial cells promotes lung injury and anti-platelet treatment improves organ damage and survival [146].

The severe acute respiratory syndrome coronavirus 2 (SARS-CoV-2) is a single-stranded RNA betacoronavirus that causes the ongoing coronavirus disease 2019 (COVID-19) pandemic. COVID-19-related ARDS is associated with severe pneumonia, thrombotic and systemic complications and high mortality rates. The role of platelets in SARS-CoV-2 infections has already been intensively investigated in the shortest possible time (Figure 4) [147].

In a study from Manne et al., extensive RNA sequencing revealed changes in the platelet transcriptome upon SARS-CoV-2 infection. Pathways elevated in patients with COVID-19 included protein ubiquitination, antigen presentation and mitochondrial dysfunction. Interferon-induced transmembrane protein 3 was significantly upregulated in COVID-19 patients, while the expression of the putative SARS-CoV-2 binding receptor angiotensin-converting enzyme 2 (ACE2) was not identified. Interestingly, a study from Koupenova et al. describes the rapid internalization and digestion of SARS-CoV-2 virions via EV-transfer. Internalization resulted in rapid cell death, as indicated by an increase in apoptotic markers [148]. Platelet counts, mean platelet volume and morphology were unaffected by SARS-CoV-2 infection. However, TPO plasma levels, CD62P expression, platelet-leukocyte aggregates, platelet aggregation and the MAPK-signaling pathway were significantly increased in COVID-19 patients [141]. These findings are supported by a study from Taus et al. showing that SARS-CoV-2 infection results in increased platelet-leukocyte aggregates. Surface CD62P on platelets derived from COVID-19 patients was not affected on either resting or activated platelets. However, COVID-19 platelets released larger amounts of cytokines, chemokines and growth factors upon stimulation, indicating pro-inflammatory priming of platelets during SARS-CoV-2 infections [149]. COVID-19 patients depict elevated plasma levels of platelet-derived PF4/RANTES, TF and NET-indicating MPO-DNA complexes, which correlated with disease severity. As already seen in previously mentioned infections, NETs also promote the pathogenesis of COVID-19. In this context, NETs trigger immunothrombosis and, thus, contribute to prothrombotic complications during SARS-CoV-2 infections [150,151]. Immune dysregulation and cytokine storms during COVID-19 contribute to platelet activation and platelet-leukocyte aggregate formation. Not only neutrophil- but also monocyte-platelet aggregates are increased during severe SARS-CoV-2 infections and are associated with CD62P- and GPIIbIIIa-dependent TF expression by monocytes [152]. TF-harboring cells shed high amounts of EVs from their surfaces. EVs, in turn, are associated with increased plasmatic thrombin generation and thromboembolic events, with endothelial cytokine release and leukocyte activation in COVID-19 patients [153].

## 4. Conclusions

Over the past decades, steadily increasing amounts of data have been published implicating that platelets do not only play a role in coagulation but also participate in pro-inflammatory as well as pro-resolution processes. Their impressive multifariousness as part of the immune response ranges from the release of various soluble mediators to a physical crosstalk with multiple immune cells. This is particularly interesting when taking the relevance of platelets for a variety of inflammatory diseases, such as ARDS and sepsis, into account. Gaining a better understanding of the platelets’ role in processes that govern the response to tissue damage and pathogen invasion will promote the development of novel therapeutic interventions.

## Figures and Tables

**Figure 1 cells-11-01957-f001:**
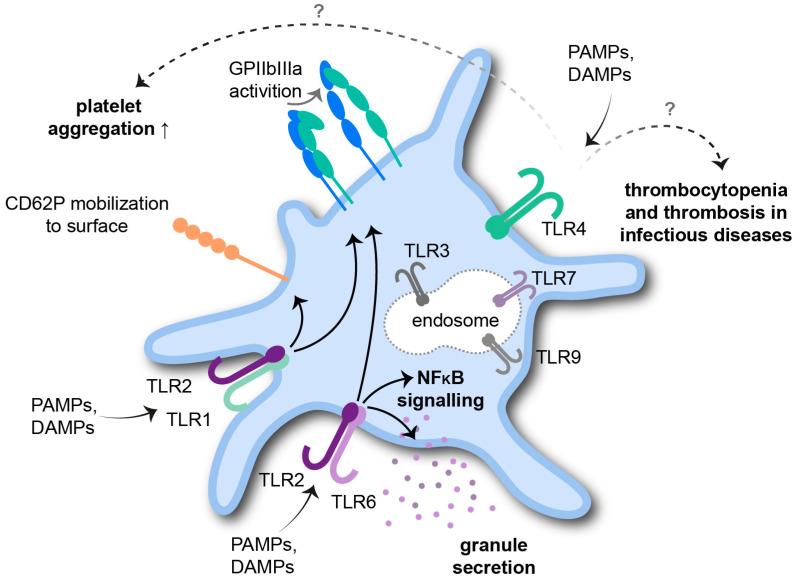
Platelet Toll-like receptors (TLRs). Platelets present functional TLR1, TLR2, TLR3, TLR4, TLR6, TLR7 and TLR9 for the recognition of invading pathogens and endogenous danger signals. Activation via pathogen-associated molecular patterns (PAMPs) and danger-associated molecular patterns (DAMPs) triggers downstream signaling pathways in platelets essential for an adequate immune response.

**Figure 2 cells-11-01957-f002:**
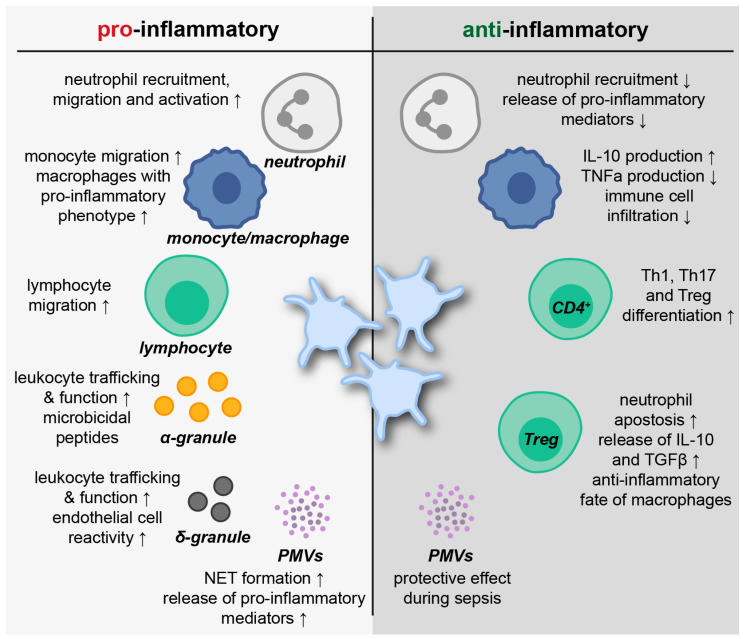
Platelet-leukocyte interactions in pro- and anti-inflammatory processes. The interaction of platelets with leukocytes via direct cell–cell contact and soluble mediators affects the immune response in multiple ways. PMVs = platelet-derived microvesicles; TNFα = tumor necrosis factor α; TGFβ = transforming growth factor β; Th = T helper cell; Treg = regulatory T cell.

**Figure 3 cells-11-01957-f003:**
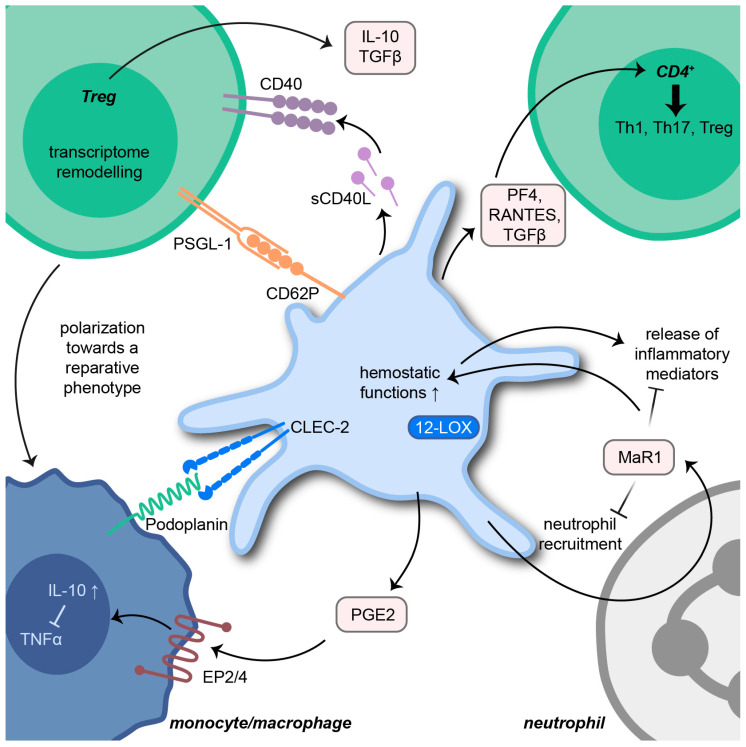
Platelets in inflammatory resolution processes. Interaction of platelets with different immune cells results in the production and release of anti-inflammatory mediators, which promote pro-resolving immune processes. Platelet-derived 12-lipoxygenase (12-LOX) initiates the synthesis of maresin 1 (MaR1), a pro-resolving protein inhibiting the release of pro-inflammatory mediators. Prostaglandin E2 (PGE2) increases the production of the anti-inflammatory cytokine interleukin (IL)-10 by monocytes/macrophages and, thereby, prevents the expression of tumor necrosis factor α (TNFα). Monocyte infiltration is further reduced by platelet C-type lectin receptor 2 (CLEC-2) binding to monocytic podoplanin. Platelet-derived platelet factor 4 (PF4), regulated and normal T cell expressed and secreted (RANTES) and transforming growth factor β (TGFβ) affect the differentiation of CD4^+^ T cells in T helper (Th) and regulatory T cells (Tregs). Interaction with Tregs via sCD40L/CD40 and P-selectin glycoprotein ligand-1 (PSGL-1)/CD62P promotes Treg-dependent polarization of macrophages toward a reparative fate.

**Figure 4 cells-11-01957-f004:**
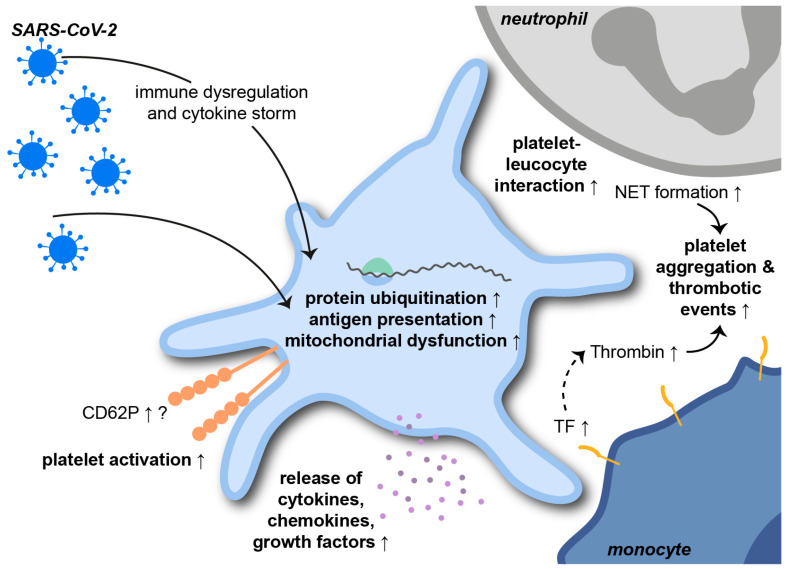
Platelets during SARS-CoV-2 infections. The severe acute respiratory syndrome coronavirus 2 (SARS-CoV-2) alters the platelet transcriptome towards enhanced protein ubiquitination, antigen presentation and mitochondrial dysfunction. SARS-CoV-2 infections are associated with enhanced platelet activation, resulting in increased platelet–leukocyte interactions and release of platelet-derived cytokines. NET = neutrophil extracellular trap; TF = tissue factor.

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
