# Peer review of "Platelets at the Crossroads of Pro-Inflammatory and Resolution Pathways during Inflammation"

_cells, 2022, doi:10.3390/cells11121957_

Round 1
Reviewer 1 Report
The present manuscript by Ludwig et al comprehensively describes the role of platelets in immunity along with their participation in immune-modulatory events. They have also listed the underlying mechanisms that drive these responses. While the article is well written, there are numerous articles available on this topic in the literature. It can also be substantially shortened. For instance, there is no need to incorporate the section on the platelet life cycle.
Author Response
The present manuscript by Ludwig et al comprehensively describes the role of platelets in immunity along with their participation in immune-modulatory events. They have also listed the underlying mechanisms that drive these responses. While the article is well written, there are numerous articles available on this topic in the literature. It can also be substantially shortened. For instance, there is no need to incorporate the section on the platelet life cycle.
Response: We would like to thank the reviewer for the positive evaluation of our manuscript. As suggested, we have shortened the text to emphasize recent findings in platelet-mediated immune responses and deleted the dedicated section on the platelet life cycle. Thus, the manuscript’s focus is accentuated and facilitates a plainer reading experience.
Reviewer 2 Report
The review entitled “Platelets at the crossroads of pro-inflammatory and resolution pathways during inflammation” adequately summarizes the indicated topic.
It is well written and structured, however has flaws regarding the content, focus, take home the message is ultimately stated poorly, and one of the prominent statements are incorrect (or worded incorrectly). I recommend major revision, however I found the manuscript overall interesting and certainly worth publishing.
Major comments:
1. The Introduction is vague and in large parts superfluous. As the article is rather long it is my recommendation to remove the historical reflections and information on thrombopoiesis, as they have no direct link to the topic. The re-worked intro to chapter “3. Immune-modulatory function of platelets” would, in my opinion, serve as a suitable Introduction after a bit of reworking.
2. I fail to understand the direct link to the topic of the entire chapter “2. Platelet life cycle”. It is interesting and well written, but unsuitable as a part of a review of platelets involvement in inflammation. My recommendation is to drastically shorten it and place as a paragraph of a following chapter, preserving only the information directly linked to the problem at hand.
3. The Conclusions are unacceptable in the current form. Please consider rewriting to provide a clear, concise message, without open-ended, imprecise statements. Please avoid using sentences such us “Investigation of these roles has provided, and will continue to provide, essential insights into a most complicated network within mammals: our immune system.” (lines 608-610) that clearly provide no information and convey no actual meaning.
4. I strongly disagree with the statement “platelets have been widely investigated in numerous research projects with a recently changing focus from traditional coagulation and wound healing studies towards their role in inflammation and inflammatory resolution processes.” Similar statements appear a couple of times more through the manuscript. I would certainly agree that the understanding of platelet function is expanding and the focus is broadening, but the research on thrombosis and haemostasis is definitely ongoing. Especially as the full understanding of haemostatic role of platelets is yet to be reached.
5. Please make sure that the minor grammatical errors do not interfere with meaning of the sentence, especially in the abstract. Currently, the sentence “These abilities make platelets key players in inflammatory diseases such as pneumonia and the acute respiratory distress syndrome, including the ongoing COVID-19 pandemic.” (lines 23 and 24) indicates the pandemic is one of the respiratory diseases.
6. The first two sections of chapter 3.5 (lines 439-466) discuss basic platelet biology, without clear link to the topic of inflammation. Please provide only pertinent information.
7. The size discrepancy between the platelet and leukocytes in the Figure 5 is extremely disconcerting and actually a bit confusing. Please make sure the platelet is at least a bit smaller than the leukocytes.
Minor comments:
· Line 58 – what is the definition of “profound” inflammation? Consider rephrasing.
· Lines 219-220 – the advertisement of another review articles seems unnecessary
· Lines 291 and 292 – I fail to see the connection of the statement “Schubert et al. revealed PMVs to be released after riboflavin/UV-induced pathogen inactivation and during extended storage of platelet concentrates.” to the main topic. As well as the general usefulness and biological meaning of this information.
· Line 310 – The sentence “During severe trauma and sepsis activated platelets enhance PMV shedding [93].” is incorrect, presumably because of a grammatical error. The correct statement is that during severe trauma and sepsis activated platelets present enhanced PMV shedding or during severe trauma and sepsis platelet activation enhances PMV shedding. Please make sure the correct meaning is prioritized above all else.
· Line 567-569 – “The severe acute respiratory syndrome coronavirus 2 (SARS-CoV-2) is a single-stranded RNA betacoronavirus and the reason for the ongoing pandemic coronavirus disease 2019 (COVID-19).” The virus causes the syndrome, both are responsible for the pandemic. Please make sure that the meaning is clear and correct.
· Lines 603 and 604 – “Over the past decades, steadily increasing amounts of data have been published implementing platelets to function outside of their classical role of coagulation, but instead within pro-inflammation and inflammatory clearance.” I believe the word is “implicating”.
Author Response
The review entitled “Platelets at the crossroads of pro-inflammatory and resolution pathways during inflammation” adequately summarizes the indicated topic. It is well written and structured, however has flaws regarding the content, focus, take home the message is ultimately stated poorly, and one of the prominent statements are incorrect (or worded incorrectly). I recommend major revision; however, I found the manuscript overall interesting and certainly worth publishing.
Response: We thank the reviewer for the insightful comments concerning our work.
Major comments:
1. The Introduction is vague and in large parts superfluous. As the article is rather long it is my recommendation to remove the historical reflections and information on thrombopoiesis, as they have no direct link to the topic. The re-worked intro to chapter “3. Immune-modulatory function of platelets” would, in my opinion, serve as a suitable Introduction after a bit of reworking.
Response: We thank the reviewer for the constructive criticism and the recommendations. As also suggested by reviewer 1, we have significantly shortened both the introduction and the platelet life cycle section and combined both parts. Now the introduction gives a summary of basic platelet facts and directs towards the role of platelets in immunological processes.
- I fail to understand the direct link to the topic of the entire chapter “2. Platelet life cycle”. It is interesting and well written, but unsuitable as a part of a review of platelets involvement in inflammation. My recommendation is to drastically shorten it and place as a paragraph of a following chapter, preserving only the information directly linked to the problem at hand.
Response: We thank the reviewer for this important suggestion. As already mentioned, we have combined this chapter with the introduction and now only report those parts considered an important background for the understanding of the following chapters (pages 1-2, lines 30-74):
“When Max Schultze and Giulio Bizzozero first described platelets as constituents of human blood and discovered their participation in wound closure in the second half of the 19th century, they might have already guessed at the potential of these tiny, abundant cells [1]. Since then, platelets have been widely investigated in numerous re-search projects with a recently expanding focus from traditional coagulation and wound healing studies towards their role in inflammation and inflammatory resolution processes.
Platelets are small, anucleate cells in the mammalian blood circulation. They are released from megakaryocytes in a process called thrombopoiesis, which is mainly driven by the acidic glycoprotein (GP) thrombopoietin (TPO) [2]. While the bone marrow remains the major site of platelet biogenesis, populations of megakaryocytes, both mature and immature, were recently found in the lungs’ extravascular space [3]. In-terestingly, lung megakaryocytes express vast amounts of toll-like receptor (TLR)2, TLR4 and major histocompatibility complex (MHC)II indicating that they are primed towards an immune-specific phenotype [4].
The final platelet concentration in human blood varies between 150-450 x 109 platelets per liter, which is maintained by a production and clearance of roughly 1011 platelets per day [5]. Platelets remain in the bloodstream for 7 to 10 days, prepared to carry out different tasks in hemostasis and immunity upon stimulation [6]. They sense and respond to a wide range of signals from the vascular endothelium and circulating blood cells by surface receptor stimulation and extensive signaling events [7]. Platelet activation results in morphological changes driven by reorganization of the cortical actin cytoskeleton from an initially disk-shaped form to a considerably enlarged fully spread platelet [8]. The physical interaction of platelets with endothelial cells, leuko-cytes and extracellular matrix components is ensured by surface integrins. Platelets primarily express β1 and β3 integrins, for instance α2β1 (GPIa/IIa) and αIIbβ3 (GPIIb/IIIa). Activation of platelets induces conformational changes of α- and β-subunits, thus enabling ligand binding and subsequent intracellular signaling cas-cades [9,10]. Since de novo protein expression in anucleate platelets is limited by the presence of only a selection of different mRNA copies, protein conversions trigger procoagulant activation, aggregation, and secretion of granule components [11]. The three major types of platelet granules - α-granules, dense granules, and lysosomal granules - are packaged with a variety of different proteins and substances capable of influencing both coagulation and inflammatory processes.
In the context of inflammation, platelets were found to capture pathogens and re-lease antimicrobial substances. Furthermore, they promote not only pro-inflammatory but also resolution processes by releasing a variety of growth factors, cytokines and chemokines and by actively interacting with neutrophils, monocytes, and lymphocytes [12]. In biomedical research the participation of platelets in immune responses is of great interest to develop novel treatment strategies for clinical use. Especially, since anti-platelet medication has previously been associated with modified immunity [13]. The coronavirus disease 2019 (COVID-19) pandemic is a prime example for ongoing intensive research on platelet behavior during severe inflammation.
This review article will provide an overview about recent findings exploring the immune modulatory functions of platelets, with a special focus on platelet-leukocyte interactions during acute inflammation and resolution in respiratory diseases.”
- The Conclusions are unacceptable in the current form. Please consider rewriting to provide a clear, concise message, without open-ended, imprecise statements. Please avoid using sentences such us “Investigation of these roles has provided, and will continue to provide, essential insights into a most complicated network within mammals: our immune system.” (lines 608-610) that clearly provide no information and convey no actual meaning.
Response: We agree with the reviewer and changed the text accordingly. The conclusion has been rewritten as suggested by the reviewer. It now provides a brief summary and a concise take-home message (pages 13-14, lines 532-541):
“Over the past decades, steadily increasing amounts of data have been published im-plicating that platelets do not only play a role in coagulation but also participate in pro-inflammatory as well as pro-resolution processes. Their impressive multifarious-ness as part of the immune response is ranging from the release of various soluble mediators to a physical crosstalk with multiple immune cells. This is especially interesting when taking the relevance of platelets for a variety of inflammatory diseases, such as ARDS and sepsis, into account. Gaining a better understanding of the platelets’ role in processes which govern the response to tissue damage and pathogen invasion will promote the development of novel therapeutic interventions.”
- I strongly disagree with the statement “platelets have been widely investigated in numerous research projects with a recently changing focus from traditional coagulation and wound healing studies towards their role in inflammation and inflammatory resolution processes.” Similar statements appear a couple of times more through the manuscript. I would certainly agree that the understanding of platelet function is expanding and the focus is broadening, but the research on thrombosis and haemostasis is definitely ongoing. Especially as the full understanding of haemostatic role of platelets is yet to be reached.
Response: We agree with the reviewer and sincerely apologize if our reporting and the focus of the review do not appreciate the ongoing research progress on the role of platelets in primary hemostasis appropriately. This is certainly not our intention. Admittedly, the chosen wording may have been misleading and has been corrected throughout the text accordingly.
Abstract, page 1, lines 11-13: “During the past decades the research focus broadened towards their participation in immune-modulatory events, including pro-inflammatory and more recently inflammatory resolution processes.”
Introduction, page 1, lines 34-36: “Since then, platelets have been widely investigated in numerous research projects with a recently expanding focus from traditional coagulation and wound healing studies towards their role in inflammation and inflammatory resolution processes.”
- Please make sure that the minor grammatical errors do not interfere with meaning of the sentence, especially in the abstract. Currently, the sentence “These abilities make platelets key players in inflammatory diseases such as pneumonia and the acute respiratory distress syndrome, including the ongoing COVID-19 pandemic.” (lines 23 and 24) indicates the pandemic is one of the respiratory diseases.
Response: We thank the reviewer for thoroughly examining the text and have comprehensively revised the manuscript with a special focus on grammatical errors.
Abstract, page 1, lines 24-25: “These abilities make platelets key players in inflammatory diseases such as pneumonia and the acute respiratory distress syndrome, including the pandemic coronavirus disease 2019.”
- The first two sections of chapter 3.5 (lines 439-466) discuss basic platelet biology, without clear link to the topic of inflammation. Please provide only pertinent information.
Response: We thank the reviewer for pointing out this concern. However, lines 366-378 (labelling according to revised manuscript) serve as an introduction into the field of inflammatory resolution processes and is considered worth mentioning. Furthermore, lines 379-393 emphasize the need for platelet inhibitors to dampen excessive pro-inflammatory and to promote inflammatory resolution processes in a clinical setting. In our opinion, this section does not only provide basic platelet biology but emphasizes that classical platelet inhibitors “not only suppress thrombotic platelet functions but also the emergence of platelet-leukocyte aggregates” (lines 384-385) and inhibit “the release of pro-inflammatory IL-1b and IL-18 under septic conditions” (lines 391-392), which is of high importance for an adequate patient treatment. However, to further underline the link to the topic of inflammation we have included a sub-clause (page 9, lines 384-385):
“Interestingly, the application of aspirin and clopidogrel not only suppress thrombotic platelet functions but also platelet-mediated pro-inflammatory processes including the emergence of platelet-leukocyte aggregates, platelet-mediated leukocyte recruitment and ROS formation.”
- The size discrepancy between the platelet and leukocytes in the Figure 5 is extremely disconcerting and actually a bit confusing. Please make sure the platelet is at least a bit smaller than the leukocytes.
Response: We thank the reviewer for pointing out this discrepancy. The figure has been revised accordingly (see new figure 4, which used to be figure 5).
Minor comments:
· Line 58 – what is the definition of “profound” inflammation? Consider rephrasing.
Response: We corrected the mistake and apologize for the inadequate choice of words.
Page 2, lines 69-71: “The coronavirus disease 2019 (COVID-19) pandemic is a prime example for ongoing intensive research on platelet behavior during severe inflammation."
- Lines 219-220 – the advertisement of another review articles seems unnecessary.
Response: We deleted the special mention of the other review and kept the reference only.
Page 4, lines 148-150: “Certainly, the response of platelets to LPS as well as TLR4-induced interaction of platelets with leukocytes is still a matter of debate [31].”
- Lines 291 and 292 – I fail to see the connection of the statement “Schubert et al. revealed PMVs to be released after riboflavin/UV-induced pathogen inactivation and during extended storage of platelet concentrates.” to the main topic. As well as the general usefulness and biological meaning of this information.
Response: We agree with the reviewer that this additional information is irrelevant for the manuscript’s main focus. The paragraph has been removed.
- Line 310 – The sentence “During severe trauma and sepsis activated platelets enhance PMV shedding [93].” is incorrect, presumably because of a grammatical error. The correct statement is that during severe trauma and sepsis activated platelets present enhanced PMV shedding or during severe trauma and sepsis platelet activation enhances PMV shedding. Please make sure the correct meaning is prioritized above all else.
Response: We apologize for this mistake. We improved the sentence according to the reviewer’s suggestion:
Page 6, lines 238-239: “During severe trauma and sepsis activated platelets present enhanced PMV shedding.”
- Line 567-569 – “The severe acute respiratory syndrome coronavirus 2 (SARS-CoV-2) is a single-stranded RNA betacoronavirus and the reason for the ongoing pandemic coronavirus disease 2019 (COVID-19).” The virus causes the syndrome, both are responsible for the pandemic. Please make sure that the meaning is clear and correct.
Response: We have changed the end of the sentence and assume that it is clearer now.
Page 12, lines 497-499 “The severe acute respiratory syndrome coronavirus 2 (SARS-CoV-2) is a single-stranded RNA betacoronavirus which causes the ongoing coronavirus disease 2019 (COVID-19) pandemic”.
- Lines 603 and 604 – “Over the past decades, steadily increasing amounts of data have been published implementing platelets to function outside of their classical role of coagulation, but instead within pro-inflammation and inflammatory clearance.” I believe the word is “implicating”.
Response: We have corrected the mistake and apologize for the mistake:
Page 13, lines 533-535: “Over the past decades, steadily increasing amounts of data have been published implicating that platelets do not only play a role in coagulation but also participate in pro-inflammatory as well as pro-resolution processes.”
Round 2
Reviewer 2 Report
Thank you to the Authors for the corrections. I find the manuscript ok for the publication in the current form.